# Neural signatures of α2-Adrenergic agonist-induced unconsciousness and awakening by antagonist

Jesus Javier Ballesteros, Jessica Blair Briscoe, Yumiko Ishizawa*

Department of Anesthesia, Critical Care & Pain Medicine, Massachusetts General Hospital, Harvard Medical School, Boston, United States

**Abstract** How the brain dynamics change during anesthetic-induced altered states of consciousness is not completely understood. The α2-adrenergic agonists are unique. They generate unconsciousness selectively through α2-adrenergic receptors and related circuits. We studied intracortical neuronal dynamics during transitions of loss of consciousness (LOC) with the α2-adrenergic agonist dexmedetomidine and return of consciousness (ROC) in a functionally interconnecting somatosensory and ventral premotor network in non-human primates. LOC, ROC and full task performance recovery were all associated with distinct neural changes. The early recovery demonstrated characteristic intermediate dynamics distinguished by sustained high spindle activities. Awakening by the α2-adrenergic antagonist completely eliminated this intermediate state and instantaneously restored awake dynamics and the top task performance while the anesthetic was still being infused. The results suggest that instantaneous functional recovery is possible following anesthetic-induced unconsciousness and the intermediate recovery state is not a necessary path for the brain recovery.

*For correspondence:
yishizawa@mgh.harvard.edu

Competing interests: The authors declare that no competing interests exist.

## Introduction

General anesthetics are unique drugs that can induce reversible unconsciousness and have been widely used for over 170 years. However, the neurophysiological mechanisms of anesthetic-induced altered states of consciousness are not completely understood. Recent electroencephalography (EEG) and neuroimaging studies significantly advanced our understanding of the brain activity during general anesthesia. Anesthetic-induced unconsciousness is now thought to be associated with profound oscillations between brain structures (*Lewis et al., 2012*; *Purdon et al., 2013*; *Akeju et al., 2014b*; *Ishizawa et al., 2016*; *Ballesteros et al., 2020*; *Patel et al., 2020*). Moreover, we recently reported that anesthetic-induced state transitions are abrupt during propofol-induced loss of consciousness (LOC) and return of consciousness (ROC) in non-human primates (*Ishizawa et al., 2016*; *Patel et al., 2020*). Discrete metastable states have been reported during emergence from isoflurane in small animals (*Hudson et al., 2014*). Abrupt state transitions might be a fundamental manner of how the brain functions during anesthetic-induced altered states of consciousness. However, oscillatory dynamics per se during transitions and unconsciousness appear to be unique to each anesthetic agent.

Dexmedetomidine is a highly selective α2-adrenergic agonist and is unique among currently available anesthetics, most of which are known to act at multiple receptors in the central nervous system. A recent neuroimaging study suggests that dexmedetomidine significantly reduces information transfer in the local and global brain networks in humans (*Hashmi et al., 2017*), consistent with the effect of propofol (*Monti et al., 2013*). Functional disconnection between brain regions is a possible mechanism suggested for dexmedetomidine-induced unconsciousness (*Akeju et al., 2014a*; *Song et al., 2017*). EEG changes under dexmedetomidine anesthesia are distinctive, with increased

slow-delta oscillations across the scalp and increased frontal spindle oscillations, and closely approximate the dynamics during human non-rapid eye movement (NREM) sleep (*Akeju et al., 2014b*; *Akeju et al., 2016*). However, direct recordings from neocortex, especially from functionally interconnected regions, with α2-adrenergic agonist are rare, despite its increasing role in clinical anesthesia. Moreover, dexmedetomidine is unique because a specific α2-adrenergic antagonist is used in veterinary medicine to reverse its sedative and anesthetic effects (*Scheinin et al., 1998*; *Kamibayashi and Maze, 2000*). Neuronal dynamics of awakening by the α2-adrenergic antagonist need to be studied.

Here we investigated how neuronal dynamics change during dexmedetomidine-induced LOC and ROC by directly recording from a functionally and anatomically interconnecting somatosensory (S1 and S2) and ventral premotor area (PMv) network, a well-studied corticocortical circuit, in non-human primates (*Figure 1A*; *Kurata, 1991*; *Tanné-Gariépy et al., 2002*; *de Lafuente and Romo, 2006*; *Garbarini et al., 2019*). The PMv is known to link sensation and decision-making as well as to

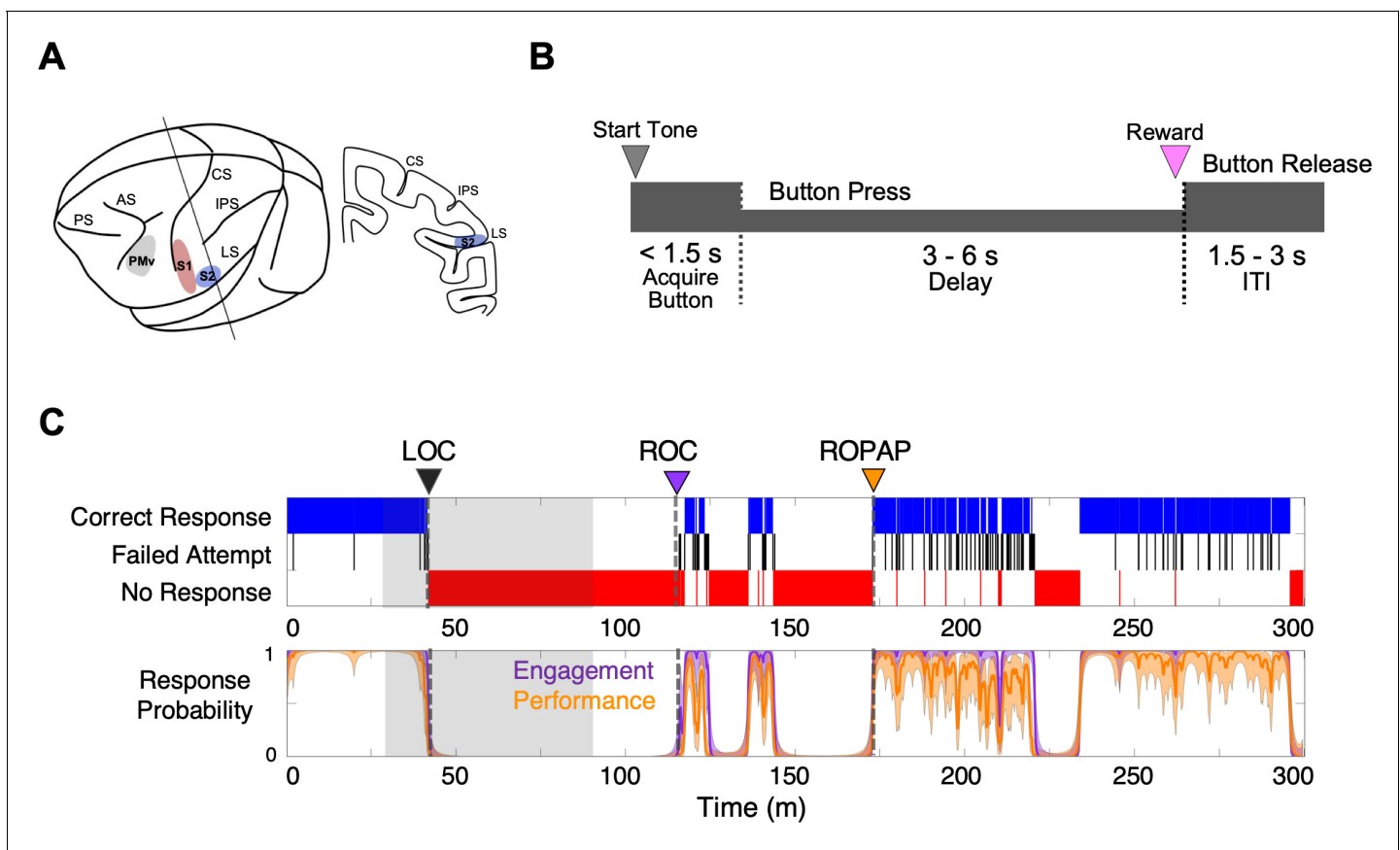

**Figure 1.** Experimental paradigm and behavioral responses. (**A**) Location of the recording sites. Neural recording was performed in the primary somatosensory cortex (S1, red), the secondary somatosensory cortex (S2, blue), and ventral premotor cortex (PMv, gray). CS, central sulcus; IPS, intraparietal sulcus; LS, lateral sulcus; AS, acuate sulcus, and PS, principal sulcus. (**B**) Behavioral task. Sequence of events during behavioral trials. After the start tone (pure tone 1000 Hz, 100 msec), the monkey initiated a trial by placing the hand (ipsilateral to the recording site) on the button in front of the animal. The animal was required to keep its hand on the button until the end of the trial in order to receive a liquid reward (correct response). The animal then had to release the button during the inter-trial interval (ITI). (**C**) Typical behavioral response during dexmedetomidine induction and emergence. Following the start of dexmedetomidine infusion, failed attempts (black) increased briefly before the animal completely lost the response. Top: The animal's trial-by-trial button response. Correct responses (blue), failed attempts (black), and no response (red). Bottom: Probability of the task engagement (correct responses and failed attempts, purple) and task performance (correct responses only, orange). LOC was defined as the time at which the probability of task engagement was decreased to less than 0.3, and ROC was defined as the first time, since being unconscious, at which the probability of task engagement was greater than 0.3. ROPAP was defined as the time at which the probability of task performance was returning to greater than 0.9 since being unconscious and remained so for at least 3 min. LOC is shown with a black arrow and dotted lines, ROC with a purple arrow and dotted lines, and ROPAP with an orange arrow and dotted lines (**C**). Dexmedetomidine was infused at 18 µg/kg/h for the first 10 min and then 4 µg/kg/h for 50 min (shaded area in **C**). The graphs represent one recording session in Monkey 1.

integrate multisensory modalities (*Rizzolatti et al., 2002*; *de Lafuente and Romo, 2005*; *de Lafuente and Romo, 2006*; *Pardo-Vazquez et al., 2008*; *Lemus et al., 2009*; *Acuña et al., 2010*; *Romo and de Lafuente, 2013*).

We used two macaque monkeys and performed multiple recording sessions with dexmedetomidine and an α2-adrenergic antagonist atipamezole, in addition to other control recording sessions, with a minimum recording interval of 2 days (*Table 1*). Dexmedetomidine was infused through a surgically implanted vascular port. In the sessions to examine the effect of the α2-adrenergic antagonist, atipamezole was administered while dexmedetomidine was still being infused. We recorded local field potentials (LFPs) and single unit activity using surgically implanted microelectrode arrays during dexmedetomidine-induced anesthesia and recovery. We defined LOC and two recovery endpoints, ROC and return of preanesthetic performance level (ROPAP), based on the probability of task engagement and task performance (*Patel et al., 2020*; *Figure 1B,C*). Task engagement indicates the probability of any response initiation by the animal, including correct responses and failed attempts, and task performance represents the probability of correct responses only (*Wong et al., 2011*; *Wong et al., 2014*; *Figure 1C*).

## Results

We successfully determined LOC, ROC and ROPAP in all the recording sessions in both animals. Interestingly, the task response behaviors appeared fluctuating, and a performing period and a non-performing period were often rapidly alternating over the course of recovery, especially during early recovery following ROC, as shown in *Figure 1A* and *Figure 2A*.

### Distinctive neural changes at α2-adrenergic agonist-induced LOC, ROC and ROPAP

We first compared LFP spectrograms from the primary and secondary somatosensory cortex (S1, S2) and ventral premotor area (PMv) during the transition from wakefulness to LOC and then through recovery. During wakefulness beta oscillations were present in both cortical regions (18–25 Hz in S1 and S2, 26–34 Hz in PMv, *Figure 2B,C,D*; *Brovelli et al., 2004*; *Haegens et al., 2011*). LOC was identified at a brief increase of the alpha power following disruption of the beta oscillations (*Figure 2B,C,D*, *Table 2*). Then the slow-delta oscillations appeared and remained dominant throughout anesthesia until ROC. ROC was associated with an abrupt diminishing of the slow-delta oscillations and an appearance of alpha oscillations (*Figure 2B,C,D*, *Table 2*). ROPAP was observed at return of the beta oscillations. The peak frequencies of the beta oscillations, however, appeared to remain significantly lower than that during wakefulness (*Figure 2F,G,H*). During an early recovery period following ROC, the slow-delta oscillations returned repeatedly when the animal was not engaged in the task and appeared to be coupled with the alpha oscillations. Both slow-delta and alpha oscillations disappeared when the beta activity returned, suggesting that two or more states were alternating until full functional recovery. We also found that dexmedetomidine did not significantly change the average firing rate in the S1 units nor in the PMv units (*Figure 2E*).

We then tested arousability during the period following LOC using a short series of non-aversive stimuli in separate sessions. Contrary to a regular session without stimuli (*Figure 2I*), we observed a brief return of task attempts following the stimuli at 3 min and 5 min after initially detected LOC (*Figure 2J*, *Table 2*). The slow-delta oscillations were disrupted, and a brief reappearance of alpha oscillations was observed in the spectrogram when the animal's task attempts returned upon the stimuli, suggesting that the slow-delta oscillations per se do not assure non-arousable state.

**Table 1.** Number of recording experiments in each animal.

| | Recording site | Number of recording sessions | | | | |
| | | Dexmedetomidine | Dexmedetomidine and antagonist | Arousability testing | No task control | Blind-folding control |
|---|---|---|---|---|---|---|
| Monkey 1 | S1, S2, PMv | 8 | 2 | 0 | 2 | 0 |
| Monkey 2 | S1, PMv | 9 | 3 | 2 | 2 | 2 |

Note: The recording sessions were performed with a minimum interval of 2 days in each animal.

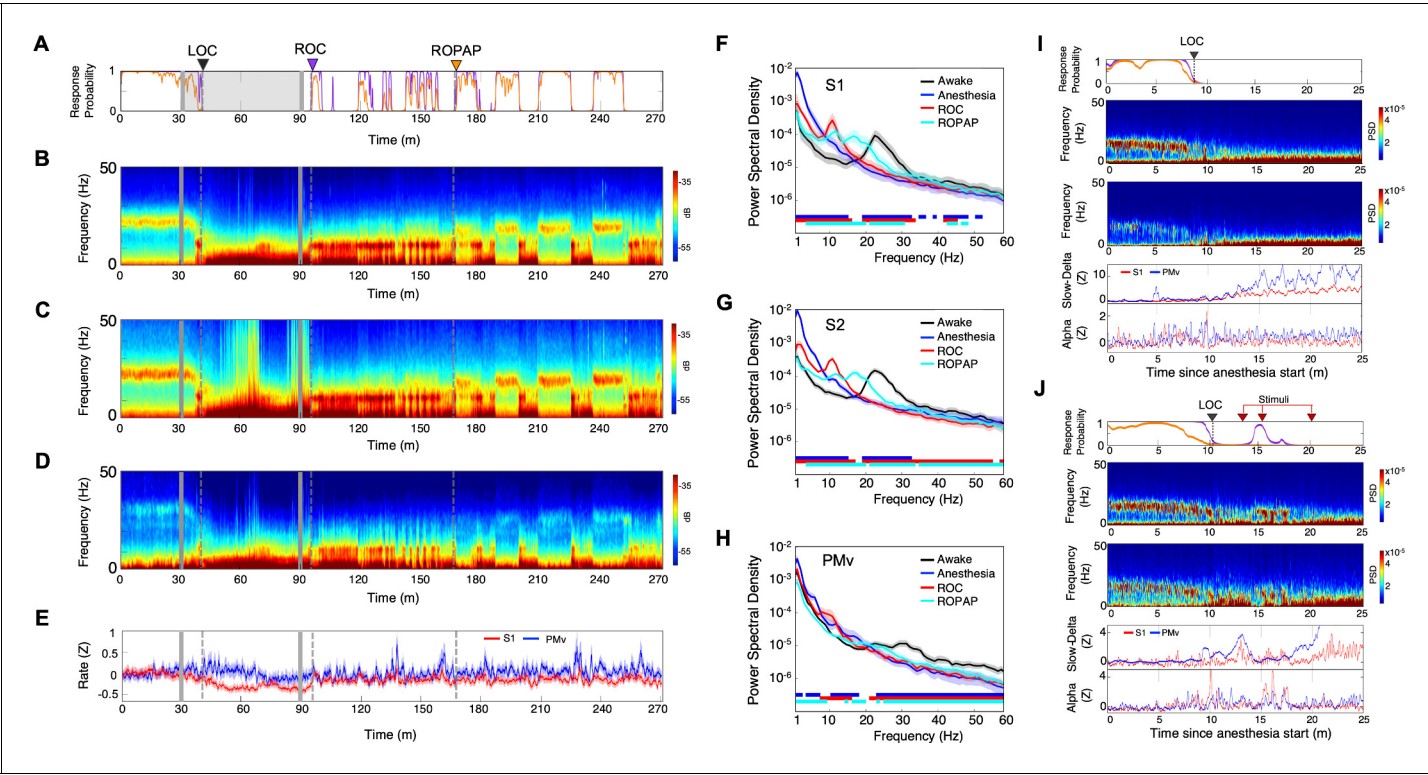

**Figure 2.** Distinctive neural changes are associated with α2-adrenergic agonist-induced LOC, ROC and ROPAP. (A) Behavioral response. Probability of the task engagement (purple) and task performance (orange). (B) Local field potentials (LFP) time-domain spectrograms in S1. (C) LFP time-domain spectrograms in S2. (D) LFP time-domain spectrograms in PMv. (E) Average baseline firing rates in S1 (red) and PMv (blue). Firing rates were normalized to pre-anesthetic values using Z-scores. No significant differences between pre-anesthetic average and any following time point were found (ANOVA and *post hoc* Bonferroni multiple comparison tests). (F–H) Averaged frequency domain power spectra in S1 (n = 9 channels, F), S2 (n = 11 channels, G) and PMv (n = 10 channels, H). Traces are the averaged Welch's power across channels with 95% confidence intervals shaded, during wakefulness (for one minute before anesthesia start, black), anesthesia (for one minute at the end of anesthetic infusion, blue), ROC (for one minute after ROC, red) and ROPAP (for one minute after ROPAP, cyan). Bottom lines represent those frequencies with significantly different values of average power density between awake and any other given condition as found by repeated measures ANOVA (ε-corrected p-value<0.05) and *post hoc* Bonferroni multiple comparison test (p-value<0.0083). (I) Behavioral response, spectrogram in S1 and PMv, the slow-delta (0.5–4 Hz) and alpha power (8–12 Hz) change during LOC. Power was normalized to pre-anesthetic values using Z-scores. (J) Behavioral response, spectrogram in S1 and PMv, the slow-delta (0.5–4 Hz) and alpha power (8–12 Hz) change during LOC with arousability testing. A series of non-aversive stimuli (ear-pulling, a loud white noise at 100 dB SPL for 5 s, and hand claps 3 times at 10 cm from face, shown with red arrows) were applied at 3, 5, and 10 min after initially detected LOC. LOC is shown with a black arrow and dotted lines, ROC with a purple arrow and dotted lines, and ROPAP with an orange arrow and dotted lines (A–E). Dexmedetomidine was infused at 18 μg/kg/h for the first 10 min and then 4 μg/kg/h for 50 min (gray lines in A–E).

**Table 2.** Characteristic oscillatory changes during dexmedetomidine-induced altered behavioral states.

| Behavioral state | Slow-Delta | Alpha | Beta | Spindle activity | |
|---|---|---|---|---|---|
| Wakefulness | minimal | absent | high | absent | *Figure 2B,C,E,F* |
| LOC | high with delay | transiently high | minimal | high | *Figure 4B* |
| Unresponsiveness | high | low | absent | high | |
| ROC | minimal* | high | absent | high | |
| ROPAP | minimal | minimal | high | minimal | |
| Arousal by external stimuli after LOC during dexmedetomidine infusion | minimal | transiently high | absent | NA | *Figure 2H* |
| Reversal by α2-adrenergic antagonist during dexmedetomidine infusion | minimal | absent | high | absent | *Figure 5B,C,E,F,G* |

Note: The characteristic changes of the frequency band power and spindle activity are summarized for each behavioral endpoint. *The slow-delta power was diminished at ROC, but frequently reappeared through early recovery until ROPAP.

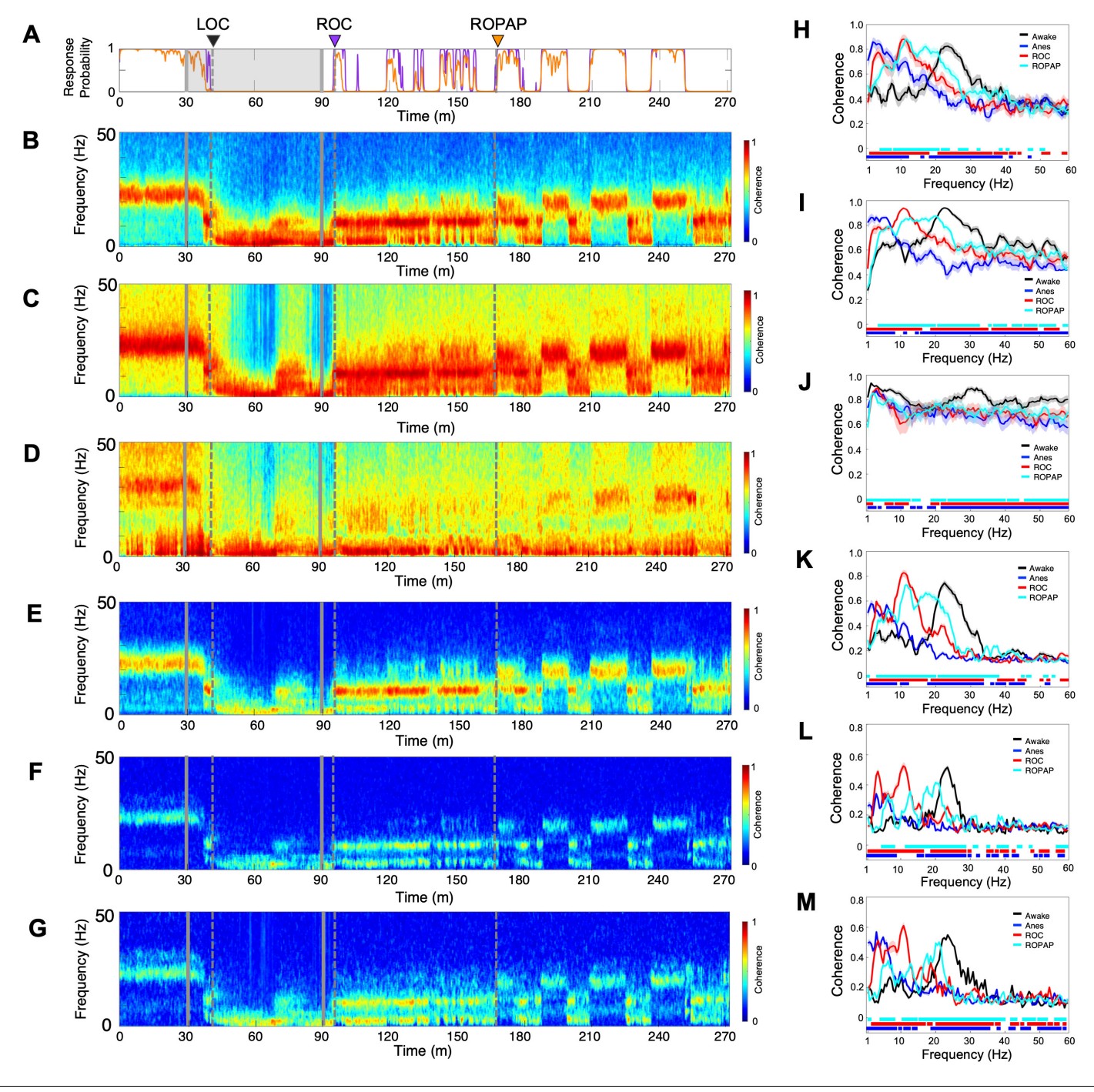

**Figure 3.** Oscillatory dynamics are inter-regionally coherent through α2-adrenergic agonist-induced anesthesia and recovery. (**A**) Behavioral response. Probability of the task engagement (purple) and task performance (orange). (**B–D**). Local field potentials (LFP) time-domain coherogram within S1 (**B**), S2 (**C**) and PMv (**D**). (**E–G**). Inter-regional LFP time-domain coherogram between S1 and S2 (**E**), S1 and PMv (**F**), and S2 and PMv (**G**). (**H–M**). Averaged frequency-domain coherence within S1 (n = 36 channel pairs, **H**), S2 (n = 55 channel pairs, **I**) and PMv (n = 45 channel pairs, **J**), and between S1-S2 (n = 99 channel pairs, **K**), S1-PMv (n = 90 channel pairs, **L**) and, S2-PMv (n = 110 channel pairs, **M**), at awake (for one minute before anesthesia start, black), anesthesia (for one minute before the end of anesthesia infusion, blue), ROC (for one minute after ROC, red) and ROPAP (for one minute after ROPAP, cyan). Coherence was averaged for a 1 min period across all pair-wise channel for each epoch. Average values are shown with shaded 95% confidence intervals. Bottom lines represent those frequencies with significantly different values of average coherence between awake and any other given condition as found by repeated measures ANOVA (ε-corrected p-value<0.05) and *post hoc* Bonferroni multiple comparison test (p-value<0.0083). *Figure 3 continued on next page*

*Figure 3 continued*

LOC is shown with a black arrow and dotted lines, ROC with a purple arrow and dotted lines, and ROPAP with an orange arrow and dotted lines (**A–G**). Dexmedetomidine was infused at 18 µg/kg/h for the first 10 min and then 4 µg/kg/h for 50 min (gray lines in **A–G**).

Arousability was not observed by the same stimuli at 10 min after the initial LOC and there was no change in the oscillatory dynamics (*Figure 2J*).

We next investigated how dexmedetomidine affects communication across S1, S2 and PMv by examining both local and regional coherence changes. We found that beta oscillations were strongly coherent locally as well as inter-regionally between S1, S2 and PMv during awake task performance (*Figure 3A–G*). However, following the start of dexmedetomidine infusion, while the animal was still performing the task, these coherent beta oscillations appeared to be disrupted. Subsequent dynamics characterized by a brief appearance of alpha oscillations at LOC and evolving slow-delta oscillations seemed to be all coherent locally and inter-regionally (*Figure 3A–G*). During recovery, the alpha and slow-delta oscillations and the beta oscillations were also coherent inter-regionally, suggesting the network is coherent throughout anesthesia and recovery. Coherence of the beta oscillations appeared to have recovered to the awake level locally and inter-regionally at ROPAP, but the peak beta frequencies at ROPAP were still significantly lower than the awake level (*Figure 3H–M*).

There was no comparable neurophysiological change at the loss of response or at the return of response in alert behaving animals without anesthetic (*Ishizawa et al., 2016*), suggesting that behavioral changes due to satiety or motivation are unlikely to be associated with the neurophysiological changes observed during dexmedetomidine-induced LOC or ROC.

## Spindle activity was highest during an early recovery period

We also investigated spindle oscillations during dexmedetomidine anesthesia and recovery. Spindle density was analyzed in the alpha and low beta frequencies between 9 and 17 Hz over the course of behavioral changes with dexmedetomidine (*Kam et al., 2019*; *Figure 4A,B*). We found that spindle activity emerged prior to LOC and the spindle density initially peaked at LOC (*Figure 4B*, *Table 2*). Spindle activity was present through dexmedetomidine-induced unconsciousness, and then further increased upon the end of anesthetic infusion and through ROC. Spindle activity was higher when the animal was performing the task prior to ROPAP, as compared to the performing period after ROPAP where the spindles were nearly completely diminished. Moreover, the spindle activity appeared to increase at the behavioral transitions between a task responding period and a non-responding period. The power of alpha frequencies seemed to be correlating with the spindle activity during anesthesia and recovery (*Figure 4C*). Spindle characteristics, including density, duration and peak frequency, were largely similar between under anesthesia and during a non-performing period after ROPAP (*Figure 4D,E,F*). However, in S1 the spindle density was statistically significantly higher during the non-performing period after ROPAP than under anesthesia, and in S2 the density was lower during the non-performing period after ROPAP than under anesthesia (*Figure 4D*). Spindle duration was significantly longer during the non-performing period after ROPAP than under anesthesia (*Figure 4E*).

## α2-adrenergic antagonist immediately restored awake dynamics and top task performance without intermediate stages

α2-adrenergic antagonist atipamezole induced an instant return of the top task performance while dexmedetomidine was still being infused (*Figure 5A*). Concurrently, oscillatory dynamics demonstrated discontinuous return of the robust beta oscillations at the frequencies that were shown during wakefulness (*Figure 5B,C,F,G*, *Table 2*). These beta oscillations were inter-regionally coherent (*Figure 5D,H*). In fact, the line coherogram indicated that the beta oscillations were significantly more coherent inter-regionally after α2-antagonist administration than during wakefulness (*Figure 5H*). The spindle activity was abruptly diminished upon the antagonist administration without showing its increase prior to ROC as observed during recovery without antagonist (*Figure 5E*).

To further quantify complex spectral dynamics during recovery from dexmedetomidine-induced unconsciousness, we investigated a three-dimensional (3D) state space using principal components analysis for three LFP spectral amplitude ratios (*Gervasoni et al., 2004*; *Hudson et al., 2014*). A

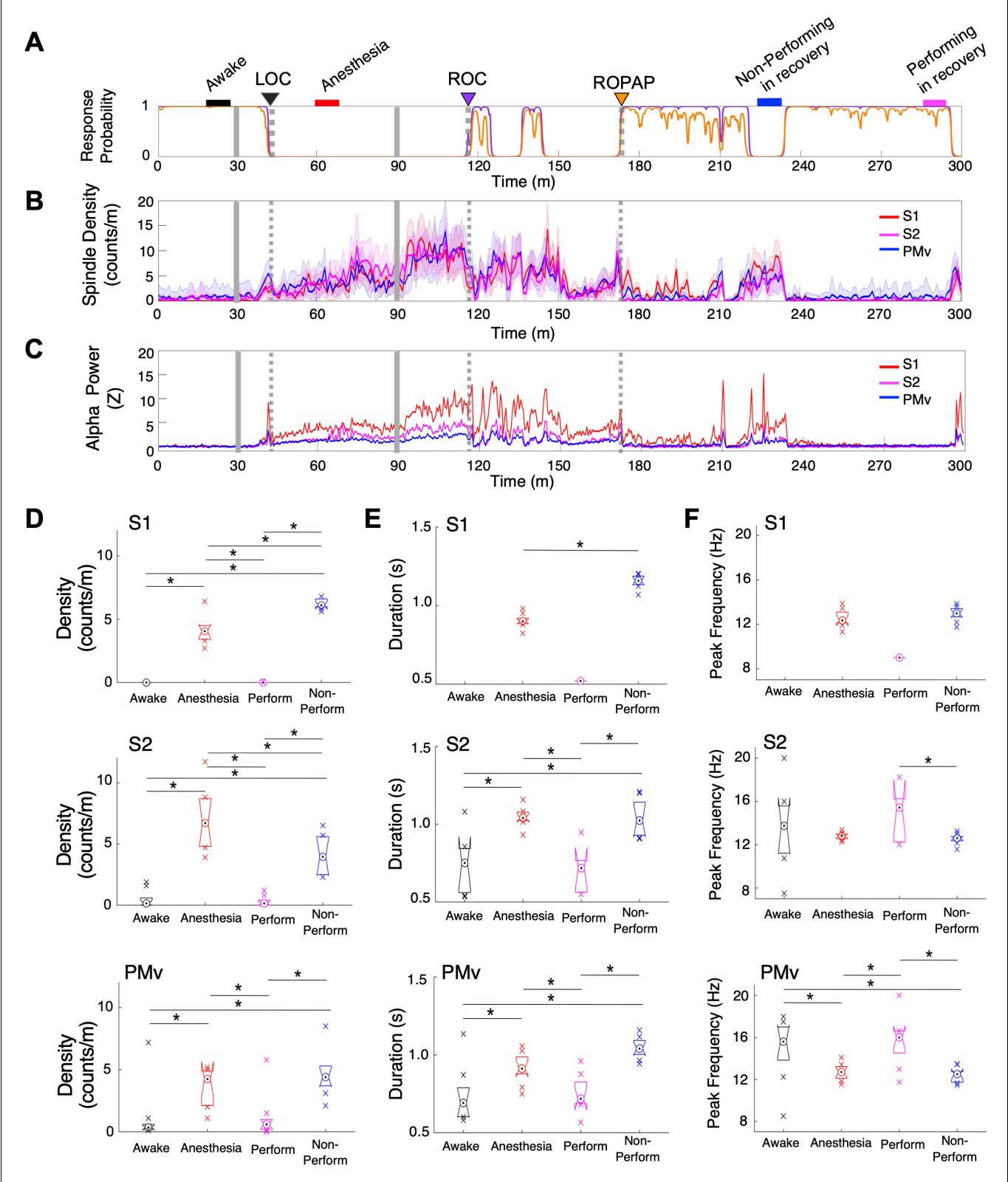

**Figure 4.** Spindle activity increases during early recovery following α2-adrenergic agonist-induced anesthesia. (A) Behavioral response. Probability of the task engagement (purple) and task performance (orange). (B) Spindle density (counts/min) in 9–17 Hz in S1 (red trace), S2 (magenta trace) and PMv (blue trace). (C) Normalized power of alpha frequency (8–12 Hz) in S1 (red trace), S2 (magenta trace) and PMv (blue trace). (D–F). Spindle characteristics in S1 (top plots), S2 (middle plots) and PMv (bottom plots) of density (D), duration (E), and peak frequency (F). Box plots represent the median with 25th

*Figure 4 continued on next page*

*Figure 4 continued*

and 75[th] percentiles and single points are all values beyond. Asterisks indicate statistically significant difference (two-sided unpaired t-test, p<0.01). Comparisons were made between the 10 min periods: awake (for the last 10 min of wakefulness before anesthesia start, black bar); anesthesia (for 10 min during anesthetic infusion, red bar); non-performing after ROPAP (for 10 min of the non-performing period, blue bar); and performing after ROPAP (for the last 10 min of the performing period, pink bar) shown in **A**. LOC is shown with a black arrow and dotted lines, ROC with a purple arrow and dotted lines, and ROPAP with an orange arrow and dotted lines (**A–C**). Dexmedetomidine was infused at 18 µg/kg/h for the first 10 min and then 4 µg/kg/h for 50 min (gray solid lines in **A–C**).

low-dimensional subspace especially allows identifying key features of the state transitions, such as density of the state and velocity of the change. We focused on the recovery dynamics without α2-antagonist (*Figure 6A,C*) versus with α2-antagonist (*Figure 6B,D*). During recovery without antagonist, we found two distinct clusters with a high-density core and a number of clouds connecting these two clusters in both S1 and PMv, possibly forming an intermediate cluster (*Figure 6* $A_{1, 2}$, $C_{1, 2}$). The intermediate area corresponded to high speed values of spontaneous trajectories (*Figure 6* $A_3$, $C_3$) and mixed task responses (*Figure 6* $A_4$, $C_4$), indicating unstable transitions. Two distinct clusters appeared to be distinguished by the animal's task performance level, high task performance versus minimum-to-zero task response. Awakening induced by α2-antagonist atipamezole demonstrated no intermediate state and two clusters were clearly separated in the state space in S1 and PMv (*Figure 6* $B_{1, 2}$, $D_{1, 2}$). The state transition from unresponsiveness to responsiveness was discontinuous and abrupt (*Figure 6* $B_3$, $D_3$). These two clusters were exclusively associated with no response and the highest performance (*Figure 6* $B_4$, $D_4$).

## Discussion

Our results demonstrate that distinctive, not gradual, neural changes were associated with the behavioral changes during α2-adrenergic agonist dexmedetomidine-induced altered states of consciousness, consistent with propofol-induced LOC and ROC (*Ishizawa et al., 2016*; *Patel et al., 2020*), even though a pharmacokinetic model assures gradual change in the anesthetic concentrations (*Patel et al., 2020*). Abrupt or non-linear state transitions are well known during natural sleep (*Saper et al., 2010*; *Stevner et al., 2019*) and in epilepsy (*Bartolomei and Naccache, 2011*). Together, these results suggest that abrupt state transitions are a fundamental manner of how the brain functions.

In the current work, a brief appearance of alpha oscillations with an increase of spindle activity was characteristic at LOC and ROC, and the slow-delta oscillations became dominant after LOC and through ROC (*Table 2*). Full task performance recovery also appeared to be associated with return of robust beta oscillations. Additionally, these characteristic oscillations were coherent locally as well as inter-regionally during anesthesia and recovery, suggesting that dexmedetomidine does not interrupt consistency of the dynamics in this cortical network. Interestingly, we have shown discrepancy between local coherence and inter-regional coherence during propofol anesthesia (*Ishizawa et al., 2016*). Especially, the high beta-gamma oscillations at propofol-induced LOC, that were highly coherent locally, were not coherent inter-regionally. Guldenmund and colleagues reported that thalamic functional connectivity within key areas of the arousal pathway was preserved during dexmedetomidine-induced unresponsiveness, but was significantly reduced during propofol anesthesia (*Guldenmund et al., 2017*). Together, these results suggest that dexmedetomidine preserves inter-regional continuity across frequencies through the state changes while propofol interrupts the communication during the state transition. The results also suggest that the distinctive neural changes in the cortical dynamics induced by α2-adrenergic agonist are likely driven by remote areas, such as subcortical regions.

Recovery from dexmedetomidine shows a characteristic intermediate state with an increase in the alpha power and high spindle density, especially during early recovery following the end of anesthetic infusion through ROC. The oscillatory dynamics of this intermediate state resembles the state of NREM and REM sleep (*Prerau et al., 2017*) and is finally replaced with coherent beta oscillations characteristic to wakefulness. Using pharmacogenetic techniques in mice, Zhang and colleagues reported that sleep-like sedation with α2-adrenergic agonist is mediated through the preoptic hypothalamic area, while loss of righting reflex, considered a surrogate for unconsciousness, requires

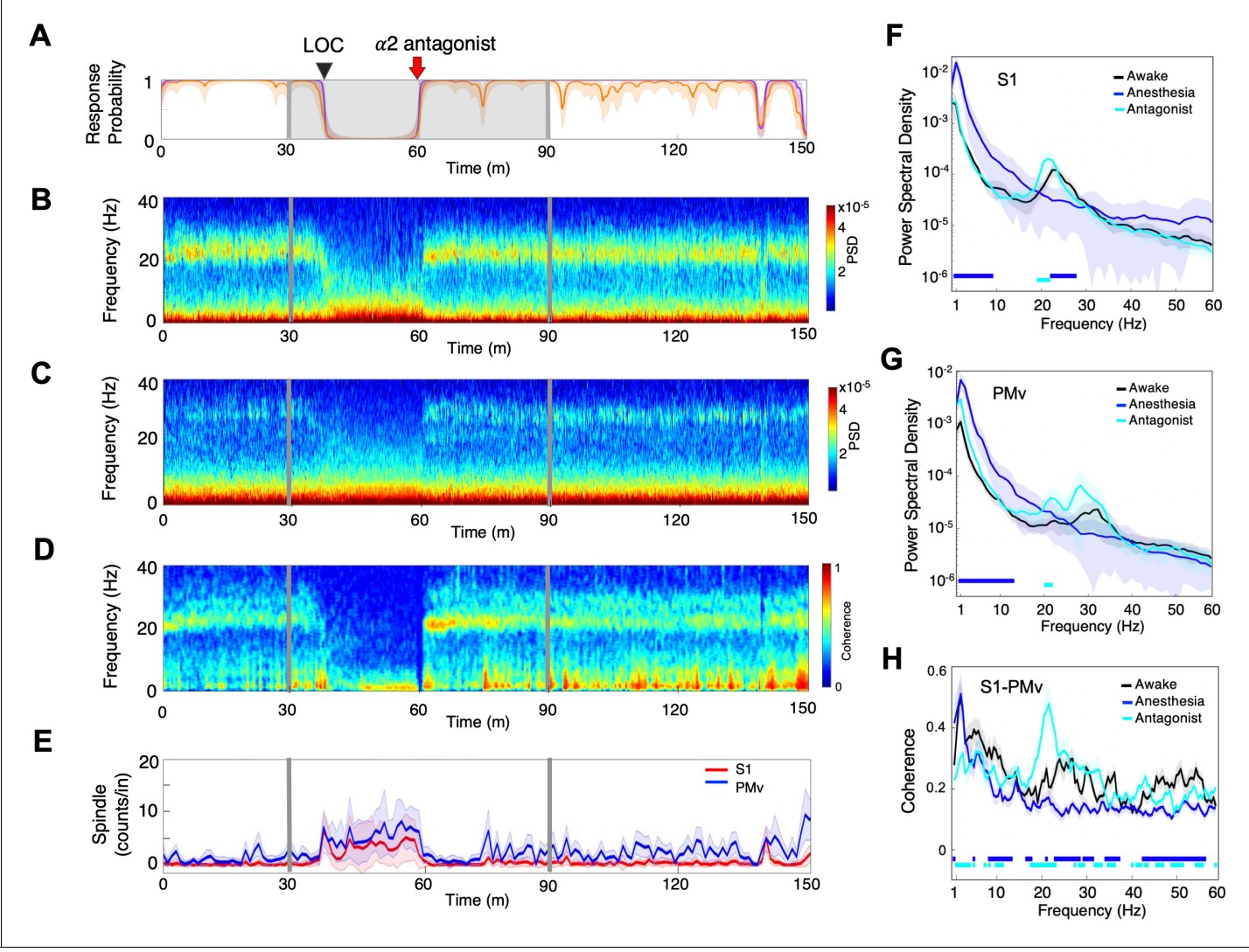

**Figure 5.** α2-adrenergic antagonist immediately restores top task performance and awake dynamics. (**A**) Behavioral response. Probability of the task engagement (purple) and task performance (orange). (**B**) Local field potentials (LFP) time-domain spectrogram in S1. (**C**) LFP time-domain spectrogram in PMv. (**D**) LFP time-domain coherogram between S1 and PMv. (**E**) Spindle density (counts/min) in 9–17 Hz in S1 (red trace) and PMv (blue trace). (**F**) Averaged frequency-domain power spectra in S1 (n = 10 channels), (**G**) Averaged frequency-domain power spectra in PMv (n = 11 channels). Traces are the averaged Welch's power across channels with shaded 95% confidence intervals, at awake (for one minute before anesthesia start, black), during anesthesia (for one minute immediately before α2-adrenergic antagonist injection, blue) and after antagonist (for one minute immediately following α2-adrenergic antagonist injection, cyan). Bottom lines represent those frequencies with significantly different values of average power density between awake and any other given condition (repeated measures ANOVA and *post hoc* Bonferroni multiple comparison test, p-value<0.017). (**H**) Averaged frequency-domain coherence between S1 and PMv (n = 110 channel pairs). Average lines are shown with shaded 95% confidence intervals. Bottom lines represent those frequencies with significantly different values of average coherence between awake and any other given condition (Friedman's test and *post hoc* Bonferroni multiple comparison test, p-value<0.017). Coherence were averaged at awake (for one minute before anesthesia start, black), during anesthesia (for one minute immediately before antagonist injection, blue) and after antagonist (for one minute immediately following α2-adrenergic antagonist injection, cyan). LOC is shown with a black arrow and dotted lines (**A–D**). Dexmedetomidine was infused at 18 µg/kg/h for the first 10 min and then 4 µg/kg/h for 50 min (gray lines and shaded area in **A–D**). α2-adrenergic antagonist atipamezole 100 µg/kg was intravenously injected after 30 min of dexmedetomidine infusion (at the time of 60 min).

locus coeruleus (LC) (*Zhang et al., 2015*). Observed behavioral state changes under dexmedetomidine in the current study, including unresponsiveness and an intermediate state with fluctuating task responses, are consistent with their findings of neuroanatomical sites of the dexmedetomidine action. The α2-adrenergic agonist at high concentrations blocks the release of norepinephrine from neurons projecting from LC to the extended areas, including the cortex, the thalamus and the

arousal pathways, resulting in wide-spread slow-delta oscillations, consistent with the dynamics shown in the unresponsive state in our study. When the α2-adrenergic agonist concentration is decreasing following the end of anesthetic infusion, the blockade of norepinephrine release may be limited to the preoptic area, which activates inhibitory GABAergic projections in the arousal centers (*Zhang et al., 2015*), similar to the effects of GABAergic general anesthetic agents. Increasing GABA$_A$ conductance leads to synchronous alpha-activity in the thalamocortical loops (*Ching et al., 2010*), consistent with our finding of inter-regionally coherent alpha oscillations during the intermediate state in early recovery from dexmedetomidine.

Although the spindle characteristics have been shown to vary across cortical regions during sleep (*Takeuchi et al., 2016*), characteristic features of the observed spindle activities with dexmedetomidine seem to closely approximate the spindles during NREM sleep in humans and non-human primates (*Nir et al., 2011*; *Takeuchi et al., 2016*), suggesting common mechanisms. In the current work, we found sustained high spindle activity following the end of anesthetic infusion. The spindle activity during this early recovery period was often higher than the one during dexmedetomidine infusion. These results suggest that, as compared to dexmedetomidine-induced anesthesia, dexmedetomidine-induced sedation or recovery from anesthesia may be neurophysiologically similar to NREM sleep. This is also consistent with the finding that dexmedetomidine activates endogenous NREM sleep-promoting pathways (*Nelson et al., 2003*; *Yu et al., 2018*). Moreover, we found a transient increase in the spindle activity at behavioral transitions, including LOC and task response shifts during recovery. The spindles are thought to increase an arousability threshold since burst firing of thalamocortical cells during spindles blocks external stimuli (*Wimmer et al., 2012*; *Astori et al., 2013*), and the spindle increase at LOC and during unresponsiveness can be explained by this possible sleep protection mechanism. Interestingly, there are some spindle peaks observed at the transition from no-response to full-response in this study. Spindles during sleep have also been reported to increase at transitional periods out of NREM sleep (*Vyazovskiy et al., 2004*), suggesting another similarity of sleep spindles and the spindles with dexmedetomidine and their possible role in the state transition. Clinically, these spindles can be useful as a sign for state changes under dexmedetomidine. When the patients are anesthetized under dexmedetomidine with minimum spindle activity, monitoring the spindle activity can be preventive against lightening the anesthetic level. When conscious or arousable sedation is appropriate, constant appearance of the high spindle activity may be a goal. Human EEG under dexmedetomidine shows dominant slow-delta and alpha-theta oscillations and the spindle activity in the frontal region following LOC (*Akeju et al., 2014b*; *Akeju et al., 2016*), consistent with our LFP findings. Together, forehead application of the EEG monitors could be modified for this new role of monitoring spindles under dexmedetomidine sedation and anesthesia.

Further, our results demonstrate for the first time that the use of an α2-adrenergic antagonist instantly restores awake cortical dynamics and concurrent high level of task performance. Previously, pioneer clinical studies demonstrated that an α2-adrenergic antagonist, atipamezole, reversed sedative and sympatholytic effects of dexmedetomidine in humans (*Karhuvaara et al., 1991*; *Aho et al., 1993*; *Scheinin et al., 1998*). Our simultaneous intracortical neurophysiology recordings and behavioral measurements suggest an abrupt transition by the α2-adrenergic antagonist, and the awakening is neither gradual nor multi-step. The 3D state space dynamics clearly characterizes a discontinuous shift upon antagonist administration without transitioning through intermediate states. During emergence without antagonist, we found diffuse clouds or possible clusters connecting between unresponsiveness and a performing state shown in *Figure 6*. This intermediate state corresponds to spectrographic signatures, such as an increase in the alpha power and the spindle activities. Hudson and colleagues first reported multiple metastable states during recovery from isoflurane anesthesia in rodents (*Hudson et al., 2014*). Proekt and Hudson further delineated stochastic dynamics of neuronal states during anesthetic recovery and explained that anesthetic recovery can be independent from pharmacokinetics (*Proekt and Hudson, 2018*). Our results with dexmedetomidine are consistent with the proposed multi-state stochastic recovery theories. However, our results also demonstrate that the probabilistic process of anesthetic recovery can be shut down by an overwhelming pharmacological effect of α2-adrenergic antagonist at the receptor level and that the intermediate recovery states can be totally bypassed. Moreover, our results support that the brain is capable to switch dynamics in an on-off manner and the expected behavioral performance can accompany without delay.

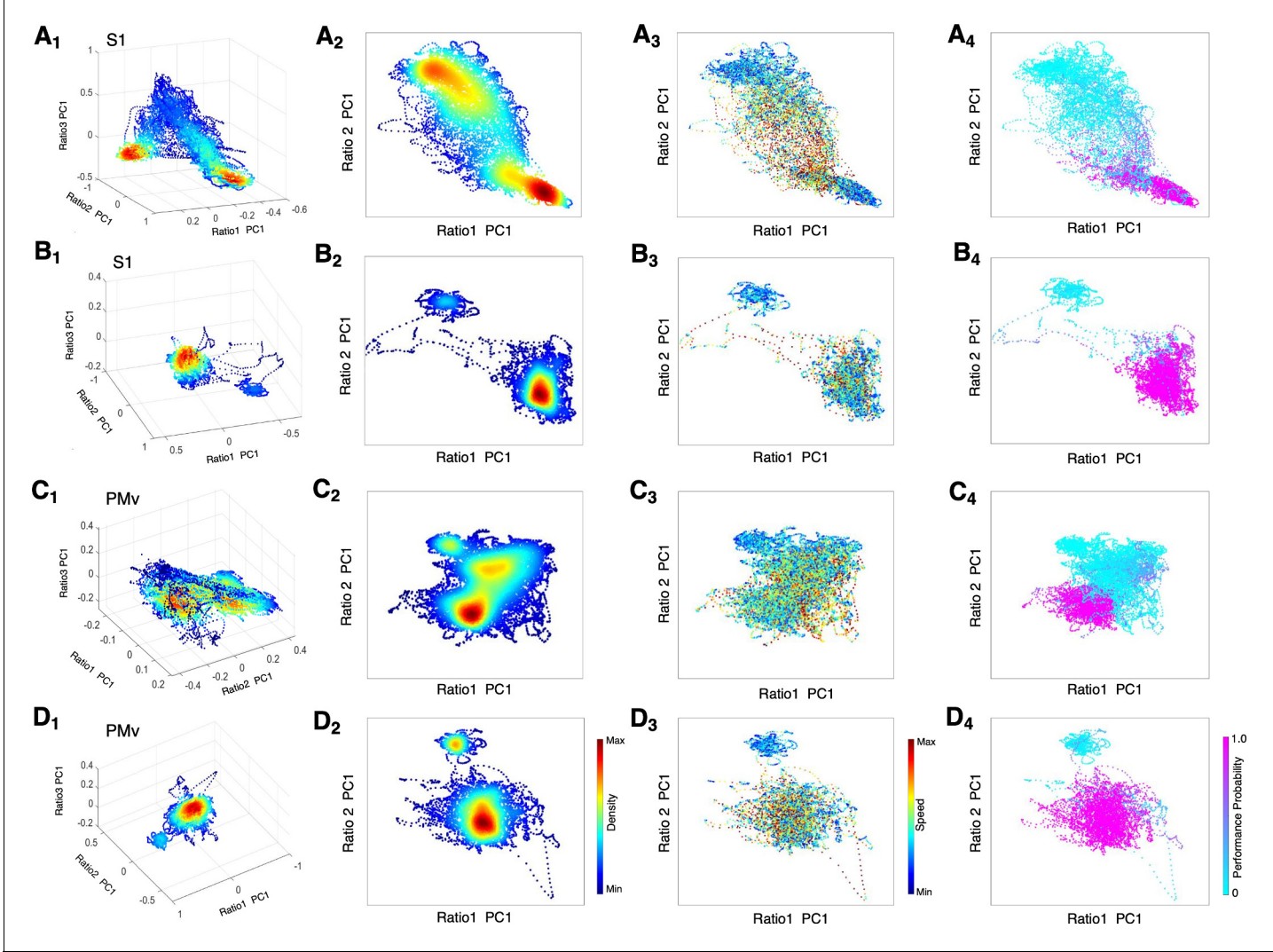

**Figure 6.** State space characterizes recovery through intermediate state without α2-antagonist and instant awakening with α2-antagonist. (A1-D1) Density plots in 3D during anesthesia and recovery without antagonist in S1 ($A_1$) and PMv ($C_1$) and with antagonist in S1 ($B_1$) and PMv ($D_1$). The density heat-map was calculated with a kernel density estimator from the scatter data created using the first principal component from the three spectral ratios described in the methods. (A2-D2) Density plots in 2D during anesthesia and recovery without antagonist in S1 ($A_2$) and PMv ($C_2$) and with antagonist in S1 ($B_2$) and PMv ($D_2$). ($A_3$-$D_3$) Speed plots during anesthesia and recovery without antagonist in S1 ($A_3$) and PMv ($C_3$) and with antagonist in S1 ($B_3$) and PMv ($D_3$). The speed was calculated as the Euclidean distance between consecutive points and is represented as heat-map. ($A_4$-$D_4$) Task performance during anesthesia and recovery without antagonist in S1 ($A_4$) and PMv ($C_4$) and with antagonist in S1 ($B_4$) and PMv ($D_4$). The plots are color-coded according to the task performance probability. The data were analyzed for the period between LOC and the end of recording session. Each dot corresponds to a 1 s window for all plots. Dexmedetomidine was infused at 18 µg/kg/h for the first 10 min and then 4 µg/kg/h for 50 min in all sessions. α2-adrenergic antagonist atipamezole 100 µg/kg was intravenously injected at 30 min of dexmedetomidine infusion.

How the unconscious state can be actively reversed has been investigated in coma patients due to brain injury, but with a limited degree of behavioral recovery (*Schiff et al., 2007*; *Xu et al., 2019*). In propofol-anesthetized humans, partial reversal of consciousness and the processed EEG index was shown by physostigmine (*Meuret et al., 2000*). In anesthetized rodents, activation of dopamine and norepinephrine pathways are shown to promote behavioral arousal with inhaled anesthetics (*Kenny et al., 2015*; *Taylor et al., 2016*). Gao and colleagues recently reported that activating nucleus gigantocellularis neurons in the reticular activating system elicited a high degree of awakening, including cortical, autonomic, and behavioral recovery, in both isoflurane anesthetized and hypoglycemia-induced coma rodents (*Gao et al., 2019*). In these small animal studies, the level of performance recovery is not yet clear. Clinically, emergence and postoperative neurocognitive

problems still affect patients undergoing surgery of all ages (*Lepousé et al., 2006*; *Vlajkovic and Sindjelic, 2007*; *Monk and Price, 2011*) and are thought to be associated with an overall increase in morbidity and mortality (*Monk et al., 2008*; *Steinmetz et al., 2009*; *Witlox et al., 2010*). Immediate restoration of the top task performance by an α2-adrenergic antagonist in the current study is promising and suggests its future clinical role in active awakening. Although dexmedetomidine per se has been shown to reduce emergence delirium and agitation in humans (*Kim et al., 2019*; *Shi et al., 2019*), it is possible that other anesthetics could benefit from active awakening and bypassing an intermediate recovery state where delirium and agitation are often observed.

The use of nonhuman primates has significant advantages in comparative neuroscience studies, including close homology in circuit architecture to humans and an ability to perform behavioral tasks that enable us to determine human-relevant behavioral endpoints during anesthesia-induced altered states of consciousness. However, there are limitations in this model. First, we used two animals due to the animal number limitations and randomization was not possible. It is also not possible to fully evaluate interindividual variability in the anesthetic effects on neuronal dynamics nor behavioral changes. Secondly, although we can prospectively determine the recording sites in nonhuman primates, intracortical neurophysiology recording sites are limited to a few sites and the recorded dynamics may not be seen in other areas. We studied a somatosensory and premotor cortical network since the network is functionally and anatomically interconnected (*Kurata, 1991*; *Tanné-Gariépy et al., 2002*; *de Lafuente and Romo, 2006*; *Garbarini et al., 2019*) and the premotor area is known for cognitive functions (*Rizzolatti et al., 2002*; *de Lafuente and Romo, 2005*; *de Lafuente and Romo, 2006*; *Pardo-Vazquez et al., 2008*; *Lemus et al., 2009*; *Acuña et al., 2010*; *Romo and de Lafuente, 2013*). Future studies are needed to examine whether the present results can be applied to more cognitively-relevant brain regions. In addition, we noticed that, in this hierachical sensory-premotor network, the observed neural changes are consistent and coherent between S1, S2 and PMv during dexmedetomidine-induced altered states of consciousness, suggesting that anesthetic effects may be more global than integrated in the neocortex (*Noel et al., 2019*; *Mashour et al., 2020*). These findings require testing in more cognitively-relevant regions as well.

In summary, behavioral endpoints, such as LOC, ROC and full task performance recovery ROPAP, are successfully defined during α2-adrenergic agonist-induced altered states of consciousness in non-human primates, and are all associated with a distinctive change in the cortical dynamics, consistent with the abrupt state transitions during propofol anesthesia and recovery (*Ishizawa et al., 2016*; *Patel et al., 2020*). The intermediate recovery state between unresponsiveness and full performance recovery is distinguished by sustained high spindle activities with rapidly fluctuating behavioral responses. Awakening by the α2-adrenergic antagonist completely eliminates the intermediate state and discontinuously restores awake neuronal dynamics and the top task performance while dexmedetomidine was still being infused. The results suggest that instant functional recovery is possible following anesthetic-induced unconsciousness and the intermediate state is not a necessary path for the brain recovery.

## Materials and methods

All animals were handled according to the institutional standards of the National Institute of Health. Animal protocol was approved by the institutional animal care and use committee at the Massachusetts General Hospital (2006N000174). We used two adult male monkeys (*Macaca mulatta*, 10–12 kg). Prior to starting the study, a titanium head post was surgically implanted on each animal. A vascular access port was also surgically implanted in the internal jugular vein (Model CP6, Access Technologies). Once the animals had mastered the following task, prior to the recording studies, extracellular microelectrode arrays (Floating Microelectrode Arrays, MicroProbes) were implanted into the primary somatosensory cortex (S1), the secondary somatosensory cortex (S2) and ventral premotor area (PMv) through a craniotomy (*Figure 1A*). Each array (1.95 × 2.5 mm) contained 16 platinum-iridium recording microelectrodes (~0.5 Meg Ohm, 1.5–4.5 mm staggered length) separated by 400 μm. The placement of arrays was guided by the landmarks on cortical surface (*Figure 1A*) and stereotaxic coordinates (*Saleem and Logothetis, 2012*). A total of five arrays were implanted in Monkey 1 (2 arrays in S1, one in S2, and two in PMv in the left hemisphere) and four arrays in Monkey 2 (2 arrays in S1, one in S2, and one and PMv in the right hemisphere). S2 array in Monkey 2 did not provide stable signals due to unknown damage. The recording experiments were

performed after 2 weeks of recovery following the array surgery. All experiments were conducted in radio-frequency shielded recording enclosures.

The animals were trained in the behavioral task shown in *Figure 1B*. After the start tone (1000 Hz, 100 msec) the animals were required to initiate each trial by holding the button located in front of the primate chair using the hand ipsilateral to the recording hemisphere. They were required to keep holding the button until the task end in order to receive a liquid reward. The monkeys were trained to perform correct response greater than 90% of the trials consistently for longer than ~1.5 hr in an alert condition. Animal's task performance during the session was monitored and simultaneously recorded using a MATLAB based behavior control system (*Asaad and Eskandar, 2008a*; *Asaad and Eskandar, 2008b*). The animal's trial-by-trial button-holding responses were used to define two metrics to allow quantification of behavioral endpoints: task engagement and task performance probability (*Patel et al., 2020*). Task engagement indicates a probability of any response initiation, including correct responses and failed attempts, and task performance represents a probability of correct responses only (*Wong et al., 2011*; *Wong et al., 2014*; *Figure 1B,C*). LOC was defined as the time at which the probability of task engagement was decreased to less than 0.3, and ROC was defined as the first time, since being unconscious, at which the probability of task engagement was greater than 0.3 (*Chemali et al., 2011*; *Mukamel et al., 2014*; *Figure 1C*). We also defined return of preanesthetic performance level (ROPAP) at which the probability of task performance was returning to greater than 0.9 since being unconscious and remained so for at least 3 min.

Dexmedetomidine was infused for total 60 min at 18 µg/kg/h for the first 10 min and then 4 µg/kg/h for 50 min through a vascular access port. The infusion rate of dexmedetomidine was determined in order to induce LOC in approximately 10 min in each animal. α2-adrenergic antagonist atipamezole (100 µg/kg) was injected through a vascular access port at 30 min of dexmedetomidine infusion while it was still being infused in four sessions and at the end of 60 min dexmedetomidine infusion in one session. No other sedatives or anesthetics were used during the experiment. The animal's heart rate and oxygen saturation were continuously monitored throughout the session (CANL-425SV-A Pulse Oximeter, Med Associates). The animals maintained greater than 94% of oxygen saturation throughout the experiments.

Neural activity was recorded continuously and simultaneously from S1, S2, and PMv through the microelectrode arrays while the animals were alert and participating in the task and throughout anesthesia and recovery. Analog data were amplified, band-pass filtered between 0.5 Hz and 8 kHz and sampled at 40 kHz (OmniPlex, Plexon). Local field potentials (LFPs) were separated by low-pass filtering at 200 Hz and down-sampled at 1 kHz. The spiking activity was obtained by high-pass filtering at 300 Hz, and a minimum threshold of three standard deviations was applied to exclude background noise from the raw voltage tracings on each channel. Action potentials were sorted using waveform principal component analysis (Offline Sorter, Plexon). The number of recordings are sumarized in *Table 1*. Recordings under dexmedetomidine anesthesia were performed eight times in Monkey 1 and 9 times in Monkey 2. In separate sessions, the recordings were performed under dexmedetomidine in the animals that were required no task performance (two sessions in Monkey 1 and 2 sessions in Monkey 2) and in the blind-folded animals that were performing the task (two sessions in Monkey 2). Arousability was tested in two sessions in Monkey 2. In addition, multiple recordings were performed in alert performing animals without anesthesia.

All LFP analyses were performed using existing and custom-written functions in MATLAB (Math-Works Inc, Natick, MA)(*Ballesteros, 2020*). Raw signals were filtered to remove 60 Hz noise by running a 2$^{nd}$ order Butterworth filter in each time direction. The resulting signals were subject to continuous multitaper spectral and channel-to-channel coherence analysis using the Chronux toolbox (*Mitra and Bokil, 2008*). Each dataset was tested for normality by Kolmogorov-Smirnov test (limiting form), conditioning the following data and statistical treatment. Sphericity was not tested but ε-corrected values did not change the interpretation of related results. To generate the spectrograms, full-length sessions were processed by running a thirty second, non-overlapping, time window, applying three tapers. The spectral lines show the average power of all channels within an array, and 95% confidence intervals (calculated by percentile method), from one-minute epoch of each condition analyzed by Welch's power spectral density estimate. Statistical analyses on the spectral change were performed comparing the awake values, frequency by frequency, versus all conditions, using repeated measures ANOVA tests (within-subject design) and post hoc Bonferroni multiple

comparisons, with a corrected significance threshold set to 0.05 divided by the number of comparisons. Channel-to-channel coherence calculations were performed running a two second, non-overlapping window. To generate coherograms we averaged all pairwise combinations within or between arrays. The average coherence for each condition was computed from one-minute epochs within each condition and it is represented as the average with 95% confidence intervals (calculated by bias-corrected bootstrap). Statistical comparisons of local and inter-regional coherence between the conditions were performed, frequency by frequency, using Friedman's tests followed by post hoc Bonferroni multiple comparisons, with a corrected significance threshold set to 0.05 divided by the number of comparisons.

Spindle detection and characterization was addressed using custom-written functions in MATLAB. We based our analysis on the methodology described by Kam and colleagues (*Kam et al., 2019*) using the FMA toolbox (*Khodagholy et al., 2017*). Briefly, we bandpass-filtered the 60 Hz noise-free LFP traces in the 9–17 Hz range. Then, we located spindle-like events by detecting their start/end (filtered signal amplitude threshold set at 4 z-scores above the baseline) and peak (threshold set at 6 z-scores). Only events with a total duration between 500 and 2500 milliseconds were further analyzed. After visual confirmation and removal of artifacts (events with peak amplitudes > 15 z-scores), we counted events located within each periods of interest: the last ten minutes before the anesthetic infusion started (awake), ten minutes during the anesthetic infusion (anesthesia), ten minutes during a performing period after ROPAP (performing in recovery) and ten minutes during a non-performing period after ROPAP (non-performing in recovery). We calculated the spindle density as number of events per minute during each period of time. The spindle peak frequency was calculated for each event's raw signal spectrogram using the Hilbert-Huang transform (*Huang et al., 2016*) and obtaining their frequency of maximum power. Data for spindle characterization were analyzed with two-sided unpaired t-tests with significant threshold set at 0.01.

To characterize the dexmedetomidine-induced brain states and their transitions, a three-dimensional space was defined using three spectral power ratios (*Gervasoni et al., 2004*; *Patel et al., 2020*). Briefly, we obtained three frequency-band spectral power ratios (Ratio1 = $power_{17-35\ Hz}$/$power_{0.5-60\ Hz}$, Ratio2 = $power_{1-8\ Hz}$/$power_{1-16\ Hz}$ and Ratio3 = $power_{9-15Hz}$/$power_{1-16\ Hz}$) from each channel. For each region and ratio, data were concatenated in a time-by-channel matrix and subjected to principal component analysis (PCA). The first principal component (PC1) explained more than 70% of the data's variance. To reduce instant variability, we smoothed the PC1 values by running a 20 s Hanning window. All figures are scatter representations of the three ratios' PC1s where each point represents one second of time. The data density was calculated with a kernel density estimator function. The speed was calculated as the Euclidean length between consecutive points. The engagement plots represent the same values of probability calculated above. The data were analyzed for the period between LOC and the end of recording session.

## Acknowledgements

We thank Emad N Eskandar, MD for performing craniotomy for electrodes implantation and supporting the project development, Tatsuo Kawai, MD for performing vascular port surgeries, Shaun R Patel, PhD for providing training in advanced data analyses, and Warren M Zapol, MD for guiding and supporting the project development.

## Additional information

### Funding

| Funder | Grant reference number | Author |
|---|---|---|
| National Institute of General Medical Sciences | 1P01GM118269 | Jesus Javier Ballesteros Yumiko Ishizawa |
| Foundation for Anesthesia Education and Research | | Yumiko Ishizawa |

The funders had no role in study design, data collection and interpretation, or the decision to submit the work for publication.

## Author contributions

Jesus Javier Ballesteros, Data curation, Formal analysis, Methodology, Writing - review and editing; Jessica Blair Briscoe, Formal analysis, Writing - review and editing; Yumiko Ishizawa, Conceptualization, Data curation, Formal analysis, Funding acquisition, Investigation, Methodology, Writing - original draft, Writing - review and editing

## Author ORCIDs

Jesus Javier Ballesteros (iD) https://orcid.org/0000-0003-2162-5117
Jessica Blair Briscoe (iD) http://orcid.org/0000-0003-3308-4215
Yumiko Ishizawa (iD) https://orcid.org/0000-0001-5393-9562

## Ethics

Animal experimentation: This study was performed in strict accordance with the recommendations in the Guide for the Care and Use of Laboratory Animals of the National Institutes of Health. All of the animals were handled according to the protocol (2006N000174) that was approved by the institutional animal care and use committee (IACUC) at the Massachusetts General Hospital.

## Decision letter and Author response

Decision letter https://doi.org/10.7554/eLife.57670.sa1
Author response https://doi.org/10.7554/eLife.57670.sa2

# Additional files

## Supplementary files

• Transparent reporting form

## Data availability

The datasets are available at Dryad https://doi.org/10.5061/dryad.98sf7m0ft.

The following dataset was generated:

| Author(s) | Year | Dataset title | Dataset URL | Database and Identifier |
|---|---|---|---|---|
| Ballesteros JJ, Briscoe J, Ishizawa Y | 2020 | Dataset01_Neural signatures of alpha2 adrenergic agonist-induced unconsciousness and awakening by antagonist | http://dx.doi.org/10.5061/dryad.98sf7m0ft | Dryad Digital Repository, 10.5061/dryad.98sf7m0ft |

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
