## [Decision Letter]

**Acceptance summary:**

This paper reports on the use of sophisticated recordings and analyses of cortical electrical activity in non-human primates and provides very new information on the changes in brain functioning that accompany sedation by an anesthetic agent called dexmedetomidine. Its originality resides in the fact that different consciousness states were studied, including the return of consciousness after the administration of a dexmedetomidine antagonist, and in the way the authors analyzed the recorded data, providing a real dissection of the recorded electrical signal. The reported results constitute a step further in the understanding of the mechanisms of anesthesia, and on the neural correlates of consciousness itself.

**Decision letter after peer review:**

Thank you for submitting your article "Neural signatures of α2 adrenergic agonist-induced unconsciousness and awakening by antagonist" for consideration by *eLife*. Your article has been reviewed by three peer reviewers, including Vincent Bonhomme as the Reviewing Editor and Reviewer #1, and the evaluation has been overseen by Richard Ivry as the Senior Editor The following individuals involved in review of your submission have agreed to reveal their identity: Adrian M Owen (Reviewer #2); Mervyn Maze (Reviewer #3).

The reviewers have discussed the reviews with one another and the Reviewing Editor has drafted this decision to help you prepare a revised submission.

Summary:

In this study, the authors have repeated their earlier local field potential spectrographic study on propofol. They record local field potentials and single unit activity in primary sensory cortex (S1) and ventral premotor area (PMv) through surgically-implanted microelectrode arrays in two adult macaque monkeys receiving a dexmedetomidine infusion. In addition to the waking state, recordings occurred during different altered brain states, including what they define as absence of consciousness (LOC = absence of a previously learned task performance), return of consciousness (ROC = return of task engagement), and return of preanesthetic performance level (ROPAP). They also performed recording after the administration of atipamezole, an antagonist of dexmedetomidine, inducing a return to awake behavior while maintaining the dexmedetomidine infusion. This last condition is an originality as compared to the previous study with propofol, which does not have a specific reversal agent.

They found spectrogram changes that were specific to LOC (disruption of beta oscillations, brief increase of alpha, and lasting slow delta, disruption of S1-MPv beta coherence, with alpha and slow-delta S1-MPv coherence), ROC (diminishing of slow-delta, appearance of alpha, alternating alpha and delta when performing task or not, S1-MPv alpha, beta, and slow-delta coherence present; same when giving stimuli during LOC and animal performing the task), and ROPAP (return of beta with lower peak frequencies than during wakefulness, inter-regionally coherent). The observed changes were not noticed during absence of task performance in the absence of anesthetic, suggesting that the observed changes are specific to the anesthetic-induced alteration of brain state. Finally, the authors modeled the brain states during recovery in the absence of antagonist, and recovery with antagonist, using a 3D space, three spectral power ratios, and principal component analysis. They found that recovery without antagonist was characterized by unstable dynamics, while the antagonist allows bypassing this intermediate state. They conclude that instantaneous functional recovery is possible following anesthetic-induced unconsciousness and that an intermediary state is not mandatory for the brain to recover.

This is an interesting and technically impressive study. The authors used a multitaper analysis to arrive at high-resolution spectral estimates. Their new findings (e.g. difference in inter-regional coherence) may provide a foundational framework to understand the pharmacological properties of propofol vs. dexmedetomidine. Their study offers new information on the brain functional changes induced by the alpha2-agonist dexmedetomidine in primates, and on transitions between those states.

There are a number of issues that require revision.

Essential revisions:

1) The paper is hard to follow in some places regarding what recordings were performed, in which order, and in which animal, all the more since the Materials and methods section is the last section. Figure 1 is informative on the study design, but does not provide enough details. The sequence of events should be better described. It is also difficult to know if the results presented in the manuscript are from one animal or both animals. Our understanding was that the results were primarily a repeated measures performed in the same animal, but we may be wrong. The authors may wish to create a table to summarize all this information (e.g., number of replicates for each animal) at a glance, for the sake of precision and clarity.

2) In the previous study a total of 29 replicates were performed on the two animals during propofol anesthesia but only 17 in the current study with dexmedetomidine. What is the reason for this discrepancy?

3) It is unclear why the data from S2 are not reported here. We understand that you reported these data separately with respect to your earlier work with propofol. However, we believe inclusion of the data from S2 would make for a more comprehensive picture and, thus increase the impact value of this paper. Unless there is a good reason to exclude these data, we recommend their inclusion.

4) The number and sites of the implanted arrays in monkeys 1 and 2 appear to be identical to monkeys E and H reported in the earlier propofol study (Ishizawa et al., 2016; and Patel et al., 2020 if reporting S2). While appreciating that the focus in the current paper should be on dexmedetomidine, we see potential value in making a direct comparison of spectral estimates between dexmedetomidine and propofol, assuming the same two animals were used for both reports.

5) The authors claim that neurocognitive problems occurring during anesthetic emergence could be avoided by appropriate antagonism. This has to be qualified by the fact that antagonists may induce withdrawal problems (and this can be the case after prolonged sedation by dexmedetomidine in the ICU), and may have short elimination half-lives to prevent risk of a return to sedation and/or fluctuating brain state after their administration (a good example is the one of naloxone to antagonize the effects of opioids). In addition, restoration of task performance by an antagonist does not necessarily mean full recovery of cognitive functions (cognitive alterations after anesthesia in human subjects may be subtle, and are certainly not limited to the ability to perform a task). We are also surprised by the authors' remarks about emergence delirium/agitation as the clinical literature is replete with reports that EA/ED is reduced by dexmedetomidine in the absence of reversal by antagonism (PMID: 31391001; PMID: 30936683). Therefore, the use of Atipamezole after dexmedetomidine to reduce EA/ED appears not to be warranted.

6) Issues that should be discussed:

a) Although multiple measures were made, only two animals were used for the study. We understand that this may be due to technical constraints, and the difficulty of implanting monkeys with electrodes and training them for task. Nonetheless, this is a limitation of the study, and should be discussed.

b) Recordings were limited to a specific somatosensory network, and the observed changes may not apply to other networks involved in consciousness phenomena. This should be discussed.

7) In this study, the ROC state is considered as a recovery state, and this is logical as it follows a period of absence of behavioral response. However, steady-state dexmedetomidine sedation, and even deep sedation, in humans is characterized by the ability to rapidly restore cooperativeness upon external stimulation. Hence, the presence of a fluctuating brain state is probably a matter of plasma or effect-site concentration, may persist, and is characteristic of dexmedetomidine sedation as compared to other sedative agents such as propofol. It has been demonstrated in human functional brain imaging studies, that at comparable levels of consciousness alteration (namely unresponsiveness), dexmedetomidine effect on the brain is characterized by a relative preservation of functional connectivity between the thalamus, the mesopontine area, and the anterior cingulate cortex, as compared to propofol sedation and sleep (see, for example, Guldenmund et al., 2017). This preservation (because concerning connectivity between the thalamus and key nodes of arousal and saliency detection) is hypothesized to be responsible for the ability of the brain to switch from one state to another. This seems to be in line with the hypothesis of the authors mention in paragraph two of the Discussion section. Can the authors link their findings to these other results?

8) With the same idea in mind, it is not clear whether the authors observed the same changes – for the studied parameters – during the ROC “up” and “down” brain states as during the changes in brain state induced by external stimulation, for all recorded parameters. Please clarify. This will be important in evaluating the idea that the ability of the brain to switch from one state to the other under dexmedetomidine sedation depends on the received dose, and that, when this occurs, the switch is abrupt.

9) For the sake of clarity and ease of reading, a table may be useful in summarizing the different findings for each parameter in a given brain state (namely: wake state, LOC, ROC, ROPAP, recovery after antagonist, and arousal by external stimulation).

10) The 3D state space model is complex and not easy to understand for readers that are not familiar with those concepts. It would be helpful to provide a short explanation on the physiological significance of the model.

11) The ability to restore responsiveness through the activation of arousal pathways (in this case, cholinergic ones) has also been described in human subjects under constant levels of propofol or sevoflurane anesthesia (see the publications of Plourde, Meuret, et al.), with corresponding electrophysiological and functional changes. It would be interesting to discuss the findings of the present study at the light of those previous studies.

12) We would like the authors to explain a little more clearly why they think spindle activity behaved as it did through the task. e.g. “We found that spindle activity emerged prior to LOC and the spindle density initially peaked at LOC. Spindle activity was present through dexmedetomidine-induced unconsciousness, and then further increased upon the end of anesthetic infusion and through ROC (Figure 4B). Spindle activity remained higher when the animal was performing the task prior to ROPAP, as compared to the performing period after ROPAP where the spindles were nearly completely diminished”. The authors do describe again this spindle activity in the Discussion, but some more nuanced explanation of why they think they see this pattern of spindle change would be welcome. To be clear, we don't have a problem with what they report, we just found it rather curious and would be interested in their further thoughts.

13) According to Aho et al. (PMID: 8100428) the optimal reversal ratio is 40:1. Using a 10 min dexmedetomidine infusion at 18 µg/kg/h and a 20 min infusion at 4 µg/kg/h, we estimate that a total of ~4 µg/kg of dexmedetomidine was infused prior to the administration of 100 µg/kg of atipamezole (ratio of 25:1). How did the authors decide on the dose of atipamezole and would they expect a different response had they used an even lower ratio of atipamezole:dexmedetomidine?

---

## [Author Response]

Essential revisions:1) The paper is hard to follow in some places regarding what recordings were performed, in which order, and in which animal, all the more since the Materials and methods section is the last section. Figure 1 is informative on the study design, but does not provide enough details. The sequence of events should be better described. It is also difficult to know if the results presented in the manuscript are from one animal or both animals. Our understanding was that the results were primarily a repeated measures performed in the same animal, but we may be wrong. The authors may wish to create a table to summarize all this information (e.g., number of replicates for each animal) at a glance, for the sake of preciseness and clarity.

We thank the reviewers for raising this point. We have added more details in the Figure 1 legend, and have created a table to summarize the number of various experimental sessions as suggested by the reviewers (Table 1 in a revised manuscript).

As shown in Table 1, recording experiments were repeated in the same animals using the same protocol. A set of results presented in each figure (e.g. behavioral response, spectrograms, and coherence) represents one session from one animal. These time-series data are not able to average across sessions or animals because the animal’s behavioral responses, such as LOC and ROC, are unique to each session. We therefore analyzed all individual sessions, but are only presenting representative sessions in the figures. Nevertheless, we averaged signals from multi-channels in the same site (the same array) in the same session. We have added this information in the figure legends (Figure 1-5).

2) In the previous study a total of 29 replicates were performed on the two animals during propofol anesthesia but only 17 in the current study with dexmedetomidine. What is the reason for this discrepancy?

In each recording session, we only used a single anesthetic agent. We performed recording with an anesthetic 1-2 times in a week with an interval of at least 2 days. Recording sessions with dexmedetomidine were performed completely separately from propofol sessions and therefore the numbers of experiments that we were able to perform were different. With dexmedetomidine, we performed more than several recording sessions in each animal shown in Table 1. We understand that the number of the sessions are sufficient for analyzing neurophysiological data and characterizing their behavioral responses.

3) It is unclear why the data from S2 are not reported here. We understand that you reported these data separately with respect to your earlier work with propofol. However, we believe inclusion of the data from S2 would make for a more comprehensive picture and, thus increase the impact value of this paper. Unless there is a good reason to exclude these data, we recommend their inclusion.

We thank the reviewers for raising these concerns. We agree that the data from S2 would make a more comprehensive picture for network dynamics. In the initial manuscript submission, we did not include the data recorded from S2 arrays because an S2 array in Monkey 2 did not provide stable signals due to unknown damage and the data were not analyzed. In the revised manuscript, we have included the data from S2 in Monkey 1 and added to Figure 2, 3 and 4.

Overall the results from S2, including spectral dynamics, coherence and spindle activity, are consistent with the S1 results. Together, the revised results suggest that dexmedetomidine preserves inter-regional continuity across frequencies in this neocortical network through the state changes: an interesting comparison with propofol that appears to interrupt the network communication.

4) The number and sites of the implanted arrays in monkeys 1 and 2 appear to be identical to monkeys E and H reported in the earlier propofol study (Ishizawa et al., 2016; and Patel et al., 2020 if reporting S2). While appreciating that the focus in the current paper should be on dexmedetomidine, we see potential value in making a direct comparison of spectral estimates between dexmedetomidine and propofol, assuming the same two animals were used for both reports.

We thank the reviewers for raising this point. As pointed out by the reviewers, we used the same two animals as in our previous study (Ishizawa et al., 2016), but used numerical numbering as suggested by an expert.

We agree that it is valuable to make a direct comparison of the spectral estimates between anesthetic agents, so that we could characterize different neural processes by different agents better. Since we have already reported the detailed dynamics for propofol-induced altered states of consciousness (Ishizawa et al., 2016 and Patel et al., 2020), we plan to perform focused analyses on the agent difference in a future manuscript.

5) The authors claim that neurocognitive problems occurring during anesthetic emergence could be avoided by appropriate antagonism. This has to be qualified by the fact that antagonists may induce withdrawal problems (and this can be the case after prolonged sedation by dexmedetomidine in the ICU), and may have short elimination half-lives to prevent risk of a return to sedation and/or fluctuating brain state after their administration (a good example is the one of naloxone to antagonize the effects of opioids). In addition, restoration of task performance by an antagonist does not necessarily mean full recovery of cognitive functions (cognitive alterations after anesthesia in human subjects may be subtle, and are certainly not limited to the ability to perform a task). We are also surprised by the authors' remarks about emergence delirium/agitation as the clinical literature is replete with reports that EA/ED is reduced by dexmedetomidine in the absence of reversal by antagonism (PMID: 31391001; PMID: 30936683). Therefore, the use of Atipamezole after dexmedetomidine to reduce EA/ED appears not to be warranted.

We thank the reviewers for raising these concerns. We also thank the reviewers for bringing these articles to our attention. We have revised this Discussion based on the following points and have cited these references.

We agree that antagonists may induce withdrawal problems, may cause over-antagonism, or may induce delayed re-sedation, as suggested by the reviewers. Those effects are unfortunately all “inappropriate” antagonism. The successful 𝛼2-adrenergic antagonism we show here is, although it is limited to this experimental condition, future possible “appropriate” antagonism to the anesthetic action by an 𝛼2 adrenergic agonist. For other general anesthetics, which actions are mediated through multiple neuronal receptors, finding “appropriate” antagonism will be significantly more challenging.

We also agree that the full task performance recovery in the current study does not necessarily indicate full recovery of cognitive functions. Task performance recovery may indicate a part of cognitive recovery, including attention and psychomotor speed to a limited extend, but not at all executive functions and memory: the functions that often affect humans following anesthesia and surgery. However, we think that our behavioral endpoint is one of the closest surrogates reported to date for recovery of cognitive functions in the animal models. Future studies with recording in more cognitively-relevant brain regions should analyze this question in detail. We have included this Discussion in the new paragraph that discussed the limitation of the study.

6) Issues that should be discussed:a) Although multiple measures were made, only two animals were used for the study. We understand that this may be due to technical constraints, and the difficulty of implanting monkeys with electrodes and training them for task. Nonetheless, this is a limitation of the study, and should be discussed.b) Recordings were limited to a specific somatosensory network, and the observed changes may not apply to other networks involved in consciousness phenomena. This should be discussed.

We thank the reviewers for raising these fundamental issues. As for representativeness, we adhere to the animal number limitations required for non-human primate research and performed as many recording sessions as animal safety and usage standards allowed. Randomization was not possible at the level of treatment given to the animals. However, we were able to perform multiple control sessions where no anesthetic drug was given as we published previously (Ishizawa et al., 2016).

We prospectively determined the areas of recording based on the scientific questions we proposed. We do agree with the reviewers that the observed changes may not be seen in other networks more involved in consciousness and cognitive functions. At the same time, the ventral premotor area (PMv) has been shown to represent some of the cognitive functions, such as subjective sensory experience and decision-making in a multiple studies (de Laufuente et al., 2006, Haegens et al., 2011, Romo and de Lafuente, 2013). In addition, in this hierachical sensory network, the observed dynamics are consistent and coherant in both S1 and PMv during the anesthetic-induced altered states of consciousness, suggesting that anesthetic effects may be more global than integrated.

We have added a new paragraph to discuss these limitations.

7) In this study, the ROC state is considered as a recovery state, and this is logical as it follows a period of absence of behavioral response. However, steady-state dexmedetomidine sedation, and even deep sedation, in humans is characterized by the ability to rapidly restore cooperativeness upon external stimulation. Hence, the presence of a fluctuating brain state is probably a matter of plasma or effect-site concentration, may persist, and is characteristic of dexmedetomidine sedation as compared to other sedative agents such as propofol. It has been demonstrated in human functional brain imaging studies, that at comparable levels of consciousness alteration (namely unresponsiveness), dexmedetomidine effect on the brain is characterized by a relative preservation of functional connectivity between the thalamus, the mesopontine area, and the anterior cingulate cortex, as compared to propofol sedation and sleep (see, for example, Guldenmund et al., 2017). This preservation (because concerning connectivity between the thalamus and key nodes of arousal and saliency detection) is hypothesized to be responsible for the ability of the brain to switch from one state to another. This seems to be in line with the hypothesis of the authors mention in paragraph two of the Discussion section. Can the authors link their findings to these other results?

We thank the reviewers for bringing this important article to our attention. We agree that preserved thalamic connectivity with key areas of arousal networks may explain the observed early recovery (return of response) following dexmedetomidine anesthesia. This may also explain the dynamics are able to fluctuate following dexmedetomidine, as opposed to propofol. However, the mechanisms of these characteristic dynamics are not entirely clear. This characteristic intermediate state could be generated by thalamus or sub-thalamic regions, which require future neurophysiology recordings in a broader network.

We have revised the discussion and have included the reference.

8) With the same idea in mind, it is not clear whether the authors observed the same changes – for the studied parameters – during the ROC “up” and “down” brain states as during the changes in brain state induced by external stimulation, for all recorded parameters. Please clarify. This will be important in evaluating the idea that the ability of the brain to switch from one state to the other under dexmedetomidine sedation depends on the received dose, and that, when this occurs, the switch is abrupt.

We thank the reviewers for pointing out this issue. We did not use external stimulation during recovery, and therefore we do not know whether the fluctuating dynamics during recovery, especially the “up” brain state, is the same as the change possibly induced by external stimulation. However, in the arousability testing, we applied a series of non-aversive external stimuli (ear-pulling, a loud white noise at 100 dB SPL for 5 sec, and hand claps 3 times at 10 cm from face) at 3, 5, and 10 minutes after initially detected LOC (Figure 2J, as compared to Figure 2I). We found a brief return of task attempts with an appearance of alpha oscillations at 3 and 5 minutes, but not at 10 minutes. The brief appearance of alpha oscillations itself seems to be consistent the alpha appearance at ROC and during early recovery (Table 2).

9) For the sake of clarity and ease of reading, a table may be useful in summarizing the different findings for each parameter in a given brain state (namely: wake state, LOC, ROC, ROPAP, recovery after antagonist, and arousal by external stimulation).

We thank the reviewers for the suggestion. We have added the table for the results of oscillatory dynamics changes during dexmedetomidine-induced altered behavioral states (Table 2 in the revised manuscript).

10) The 3D state space model is complex and not easy to understand for readers that are not familiar with those concepts. It would be helpful to provide a short explanation on the physiological significance of the model.

We thank the reviewers for raising the issue. We have added a short explanation on the physiological importance of the state space analysis in the Results section.

Briefly, as you know, the transitions between states often show continuous and dynamic changes in time, sometimes in very fast and non-unidimensional ways. We applied the dimensionality reduction of interesting frequency ratios through principal components analysis, which allows identifying key features of the state transitions, such as density of the state and velocity of the change. And we are able to overlay other analytical or behavioral measurements on top of these low dimensional subspace to observe and decide which of them may or not be significative.

11) The ability to restore responsiveness through the activation of arousal pathways (in this case, cholinergic ones) has also been described in human subjects under constant levels of propofol or sevoflurane anesthesia (see the publications of Plourde, Meuret, et al.), with corresponding electrophysiological and functional changes. It would be interesting to discuss the findings of the present study at the light of those previous studies.

We thank the reviewers for bringing this article to our attention. We have revised the Discussion and cited the reference.

12) We would like the authors to explain a little more clearly why they think spindle activity behaved as it did through the task. e.g. “We found that spindle activity emerged prior to LOC and the spindle density initially peaked at LOC. Spindle activity was present through dexmedetomidine-induced unconsciousness, and then further increased upon the end of anesthetic infusion and through ROC (Figure 4B). Spindle activity remained higher when the animal was performing the task prior to ROPAP, as compared to the performing period after ROPAP where the spindles were nearly completely diminished”. The authors do describe again this spindle activity in the Discussion, but some more nuanced explanation of why they think they see this pattern of spindle change would be welcome. To be clear, we don't have a problem with what they report, we just found it rather curious and would be interested in their further thoughts.

We thank the reviewers for bringing up this issue. We have revised the Discussion to better explain possible spindle roles based on the similarities to sleep spindles. We agree that the observed spindles activity over the course was interesting. Overall, our findings agree with the spindle’s role of sleep protection. Observed increases in the spindle activity at transitional periods from non-performing to performing during recovery is indeed interesting. This could be a new role as reported by Vyazovskiy and colleagues in small animals. However, it would be difficult to discuss the mechanistic aspects of the spindles in more detail since our data are limited to the cortical regions.

13) According to Aho et al. (PMID: 8100428) the optimal reversal ratio is 40:1. Using a 10 min dexmedetomidine infusion at 18 µg/kg/h and a 20 min infusion at 4 µg/kg/h, we estimate that a total of ~4 µg/kg of dexmedetomidine was infused prior to the administration of 100 µg/kg of atipamezole (ratio of 25:1). How did the authors decide on the dose of atipamezole and would they expect a different response had they used an even lower ratio of atipamezole:dexmedetomidine?

We thank the reviewers for pointing out this dosing issue. In the current work, we followed the dose reported in healthy humans by Scheinin et al., 1998. The authors reported that the dose of atipamezole needed to reverse the sedative effect of dexmedetomidine was 104+/-44 µ/kg. They also reported the dose dependent effect using 3 different dose (15, 50 and 150 µ/kg). We expect a different response with a lower dose of atipamezole. However, we were not able to perform dose-response experiments due to the animal use and recording schedule constraints.